# *α*-Geodesical Skew Divergence

**DOI:** 10.3390/e23050528

**Published:** 2021-04-25

**Authors:** Masanari Kimura, Hideitsu Hino

**Affiliations:** 1Department of Statistical Science, School of Multidisciplinary Sciences, The Graduate University for Advanced Studies (SOKENDAI), Kanagawa 240-0193, Japan; 2The Institute of Statistical Mathematics, Tokyo 190-0014, Japan; hino@ism.ac.jp

**Keywords:** KL-divergence, JS-divergence, skew divergence, information geometry

## Abstract

The asymmetric skew divergence smooths one of the distributions by mixing it, to a degree determined by the parameter λ, with the other distribution. Such divergence is an approximation of the KL divergence that does not require the target distribution to be absolutely continuous with respect to the source distribution. In this paper, an information geometric generalization of the skew divergence called the α-geodesical skew divergence is proposed, and its properties are studied.

## 1. Introduction

Let (X,F,μ) be a measure space where X denotes the sample space, F the σ-algebra of measurable events, and μ a positive measure. The set of the strictly positive probability measure P is defined as
(1)$$P\left\{f\left(x\right)>0\;\left(\forall x\in X\right),\;\mathrm{and}\;\int_{X}f\left(x\right)d\mu \left(x\right)=1\right\},$$
and the set of nonnegative probability measure $P_{+}$ is defined as
(2)$$P_{+}\left\{f\left(x\right)\geq 0\;\left(\forall x\in X\right),\;\mathrm{and}\;\int_{X}f\left(x\right)d\mu \left(x\right)=1\right\}.$$

Then a number of divergences that appear in statistics and information theory [1,2] are introduced.

**Definition** **1**(Kullback–Leibler divergence [3]). *The Kullback–Leibler divergence or KL-divergence $D_{KL}:P_{+}\times P\to \left[0,\infty \right]$ is defined between two Radon–Nikodym densities p and q of μ-absolutely continuous probability measures by*
(3)$$D_{KL}\left[p∥q\right]\int_{X}p\ln\frac{p}{q}d\mu .$$

KL-divergence is a measure of the difference between two probability distributions in statistics and information theory [4,5,6,7]. This is also called the relative entropy and is known not to satisfy the axiom of distance. Because the KL-divergence is asymmetric, several symmetrizations have been proposed in the literature [8,9,10].

**Definition** **2**(Jensen–Shannon divergence [8]). *The Jensen–Shannon divergence or JS-divergence $D_{JS}:P\times P\to \left[0,\infty \right)$ is defined between two Radon–Nikodym densities p and q of μ-absolutely continuous probability measures by*
(4)$$\begin{array}{cc} D_{JS}\left[p∥q\right] & :=\frac{1}{2}\left(D_{KL}\left[p∥\frac{p+q}{2}\right]+D_{KL}\left[q∥\frac{p+q}{2}\right]\right)\\ \; & =\frac{1}{2}\int_{X}\left(p\ln\frac{2p}{p+q}+q\ln\frac{2q}{p+q}\right)d\mu \\ \; & =D_{JS}\left[q∥p\right]. \end{array}$$

The JS-divergence is a symmetrized and smoothed version of the KL-divergence, and it is bounded as
(5)$$0\leq D_{JS}\left[p∥q\right]\leq \ln2.$$

This property contrasts with the fact that KL-divergence is unbounded.

**Definition** **3**(Jeffreys divergence [11])**.**
*The Jeffreys divergence $D_{J}\left[p∥q\right]:P\times P\to \left[0,\infty \right]$ is defined between two Radon–Nikodym densities p and q of μ-absolutely continuous probability measures by*
(6)$$D_{J}\left[p∥q\right]:=D_{KL}\left[p∥q\right]+D_{KL}\left[q∥p\right].$$

Such symmetrized KL-divergences have appeared in various pieces of literature [12,13,14,15,16,17,18].

For continuous distributions, the KL-divergence is known to have computational difficulty. To be more specific, if *q* takes a small value relative to *p*, the value of $D_{KL}\left[p∥q\right]$ may diverge to infinity. The simplest idea to avoid this is to use very small ϵ>0 and modify $D_{KL}\left[p∥q\right]$ as follows: $$D_{KL}^{+}\left[p∥q\right]:=\int_{X}p\ln\frac{p}{q+\epsilon }d\mu .$$

However, such an extension is unnatural in the sense that q+ϵ no longer satisfies the condition for a probability measure: $\int_{X}\epsilon +q\left(x\right)d\mu \left(x\right)\neq 1$. As a more natural way to stabilize KL-divergence, the following skew divergences have been proposed:

**Definition** **4**(Skew divergence [8,19])**.**
*The skew divergence $D_{S}^{\left(\lambda \right)}\left[p∥q\right]:P\times P\to \left[0,\infty \right]$ is defined between two Radon–Nikodym densities p and q of μ-absolutely continuous probability measures by*
(7)$$\begin{array}{cc} D_{S}^{\left(\lambda \right)}\left[p∥q\right] & :=D_{KL}\left[p∥\left(1-\lambda \right)p+\lambda q\right]\\ \; & =\int_{X}p\ln\frac{p}{(1-\lambda )p+\lambda q}d\mu , \end{array}$$
*where λ∈[0,1].*

Skew divergences have been experimentally shown to perform better in applications such as natural language processing [20,21], image recognition [22,23] and graph analysis [24,25]. In addition, there is research on quantum generalization of skew divergence [26].

The main contributions of this paper are summarized as follows:Several symmetrized divergences or skew divergences are generalized from an information geometry perspective.It is proved that the natural skew divergence for the exponential family is equivalent to the scaled KL-divergence.Several properties of geometrically generalized skew divergence are proved. Specifically, the functional space associated with the proposed divergence is shown to be a Banach space.

Implementation of the proposed divergence is available on GitHub (https://github.com/nocotan/geodesical_skew_divergence (accessed on 3 April 2021)).

## 2. *α*-Geodesical Skew Divergence

The skew divergence is generalized based on the following function.

**Definition** **5**(*f*-interpolation)**.**
*For any a,b,∈R, λ∈[0,1] and α∈R, f-interpolation is defined as*
(8)$$m_{f}^{\left(\lambda ,\alpha \right)}\left(a,b\right)=f_{\alpha }^{-1}\left(\left(1-\lambda \right)f_{\alpha }\left(a\right)+\lambda f_{\alpha }\left(b\right)\right),$$
*where*
(9)$$f_{\alpha }\left(x\right)=\left\{\begin{array}{cc} x^{\frac{1-\alpha }{2}} & \left(\alpha \neq 1\right)\\ \ln x & \left(\alpha =1\right) \end{array}\right.$$
*is the function that defines the f-mean [27].*

The *f*-mean function satisfies
$$\begin{array}{cc} \lim_{\alpha \to \infty }f_{\alpha }\left(x\right) & =\left\{\begin{array}{cc} \infty & (|x|<1),\\ 1 & (|x|=1),\\ 0 & (|x|>1), \end{array}\right.\\ \lim_{\alpha \to -\infty }f_{\alpha }\left(x\right) & =\left\{\begin{array}{cc} 0 & (|x|<1),\\ 1 & (|x|=1),\\ \infty & (|x|>1). \end{array}\right. \end{array}$$

It is easy to see that this family includes various known weighted means including the *e*-mixture and *m*-mixture for α=±1 in the literature of information geometry [28]:$$\begin{array}{cc} (\alpha =1)\;\; & \;m_{f}^{\left(\lambda ,1\right)}\left(a,b\right)=\exp\left\{\left(1-\lambda \right)\ln a+\lambda \ln b\right\}\\ (\alpha =-1)\;\; & \;m_{f}^{\left(\lambda ,-1\right)}\left(a,b\right)=\left(1-\lambda \right)a+\lambda b\\ (\alpha =0)\;\; & \;m_{f}^{\left(\lambda ,0\right)}\left(a,b\right)={\left(\left(1-\lambda \right)\sqrt{a}+\lambda \sqrt{b}\right)}^{2}\\ (\alpha =3)\;\; & \;m_{f}^{\left(\lambda ,3\right)}\left(a,b\right)=\frac{1}{\left(1-\lambda \right)\frac{1}{a}+\lambda \frac{1}{b}}\\ (\alpha =\infty )\;\; & \;m_{f}^{\left(\lambda ,\infty \right)}\left(a,b\right)=\min\left\{a,b\right\}\\ (\alpha =-\infty )\;\; & \;m_{f}^{\left(\lambda ,-\infty \right)}\left(a,b\right)=\max\left\{a,b\right\} \end{array}$$

The inverse function $f_{\alpha }^{-1}$ is convex when α∈[−1,1], and concave when α∈(−∞,−1]∪(1,∞). It is worth noting that the *f*-interpolation is a special case of the Kolmogorov–Nagumo average [29,30,31] when α is restricted in the interval [−1,1].

In order to consider the geometric meaning of this function, the notion of the statistical manifold is introduced.

### 2.1. Statistical Manifold

Let
(10)$$S=\{p_{\xi }=p\left(x;\xi \right)\in P|\xi =\left(\xi^{1},\ldots ,\xi^{n}\right)\in \Xi \}$$
be a family of probability distribution on X, where each element $p_{\xi }$ is parameterized by *n* real-valued variables $\xi =\left(\xi^{1},\ldots ,\xi^{n}\right)\in \Xi \subset R^{n}$. The set S is called a statistical model and is a subset of P. We also denote $\left(S,g_{ij}\right)$ as a statistical model equipped with the Riemannian metric $g_{ij}$. In particular, let $g_{ij}$ be the Fisher–Rao metric, which is the Riemannian metric induced from the Fisher information matrix [32].

In the rest of this paper, the abbreviations
$$\begin{array}{cc} \partial_{i} & =\partial_{\xi^{i}}=\frac{\partial }{\partial \xi^{i}},\\ \ell & =\ell_{x}\left(\xi \right)=\ln p_{\xi }\left(x\right) \end{array}$$
are used.

**Definition** **6**(Christoffel symbols)**.**
*Let $g_{ij}$ be a Riemannian metric, particularly the Fisher information matrix, then the Christoffel symbols are given by*
(11)$$\Gamma_{ij,k}=\frac{1}{2}\left(\partial_{i}g_{jk}+\partial_{j}g_{ik}-\partial_{k}g_{ij}\right),\;i,j,k=1,\ldots ,n.$$

**Definition** **7**(Levi-Civita connection)**.**
*Let g be a Fisher–Riemannian metric on S which is a 2-covariant tensor defined locally by*
$$g\left(X_{\xi },Y_{\xi }\right)=\sum_{i,j=1}^{n}g_{ij}\left(\xi \right)a^{i}\left(\xi \right)b^{j}\left(\xi \right),$$
*where $X_{\xi }=\sum_{i=1}^{n}a^{i}\left(\xi \right)\partial_{i}p_{\xi }$ and $Y_{\xi }=\sum_{i=1}^{n}b^{i}\left(\xi \right)\partial_{i}p_{\xi }$ are vector fields in the 0-representation on S. Then, its associated Levi-Civita connection $\nabla^{\left(0\right)}$ is defined by*
(12)$$g\left(\nabla_{\partial_{i}}^{\left(0\right)}\partial_{j},\partial_{k}\right)=\Gamma_{ij,k}.$$

The fact that $\nabla^{\left(0\right)}$ is a metrical connection can be written locally as
(13)$$\partial_{k}g_{ij}=\Gamma_{ki,j}+\Gamma_{kj,i}.$$

It is worth noting that the superscript α of $\nabla^{\left(\alpha \right)}$ corresponds to a parameter of the connection. Based on the above definitions, several connections parameterized by the parameter α are introduced. The case α=0 corresponds to the Levi-Civita connection induced by the Fisher metric.

**Definition** **8**($\nabla^{\left(1\right)}$-connection)**.**
*Let g be the Fisher-Riemannian metric on S, which is a 2-covariant tensor. Then, the $\nabla^{\left(1\right)}$-connection is defined by*
(14)$$g\left(\nabla_{\partial_{i}}^{\left(1\right)}\partial_{j},\partial_{k}\right)=E_{\xi }\left[\partial_{i}\partial_{j}\ell \partial_{k}\ell \right].$$
*It can also be expressed equivalently by explicitly writing as the Christoffel coefficients*
(15)$$\Gamma_{ij,k}^{\left(1\right)}\left(\xi \right)=E_{\xi }\left[\partial_{i}\partial_{j}\ell \partial_{k}\ell \right].$$


**Definition** **9**($\nabla^{\left(-1\right)}$-connection)**.**
*Let g be the Fisher–Riemannian metric on S, which is a 2-covariant tensor. Then, the $\nabla^{\left(-1\right)}$-connection is defined by*
(16)$$g\left(\nabla_{\partial_{i}}^{\left(-1\right)}\partial_{j},\partial_{k}\right)=\Gamma_{ij,k}^{\left(-1\right)}\left(\xi \right)=E_{\xi }\left[\left(\partial_{i}\partial_{j}\ell +\partial_{i}\ell \partial_{j}\ell \right)\partial_{k}\ell \right].$$

In the following, the ∇-flatness is considered with respect to the corresponding coordinates system. More details can be found in [28].

**Proposition** **1.**
*The exponential family is $\nabla^{\left(1\right)}$-flat.*


**Proposition** **2.**
*The exponential family is $\nabla^{\left(-1\right)}$-flat if and only if it is $\nabla^{\left(0\right)}$-flat.*


**Proposition** **3.**
*The mixture family is $\nabla^{\left(-1\right)}$-flat.*


**Proposition** **4.**
*The mixture family is $\nabla^{\left(1\right)}$-flat if and only if it is $\nabla^{\left(0\right)}$-flat.*


**Proposition** **5.**
*The relation between the foregoing three connections is given by*
(17)$$\nabla^{\left(0\right)}=\frac{1}{2}\left(\nabla^{\left(-1\right)}+\nabla^{\left(1\right)}\right).$$


**Proof.** It suffices to show
$$\Gamma_{ij,k}^{\left(0\right)}=\frac{1}{2}\left(\Gamma_{ij,k}^{\left(-1\right)}+\Gamma_{ij,k}^{\left(1\right)}\right).$$From the definitions of $\Gamma^{\left(-1\right)}$ and $\Gamma^{\left(1\right)}$,
$$\begin{array}{cc} \Gamma_{ij,k}^{\left(-1\right)}+\Gamma_{ij,k}^{\left(1\right)} & =E_{\xi }\left[\left(\partial_{i}\partial_{j}\ell +\partial_{i}\ell \partial_{j}\ell \right)\partial_{k}\ell \right]+E_{\xi }\left[\partial_{i}\partial_{j}\ell \partial_{k}\ell \right]\\ \; & =E_{\xi }\left[\left(2\partial_{i}\partial_{j}\ell +\partial_{i}\ell \partial_{j}\ell \right)\partial_{k}\ell \right]\\ \; & =2E_{\xi }\left[\left(\partial_{i}\partial_{j}\ell +\frac{1}{2}\partial_{i}\ell \partial_{j}\ell \right)\partial_{k}\ell \right]\\ \; & =2\Gamma_{ij,k}^{\left(0\right)}, \end{array}$$
which proves the proposition. □

The connections $\nabla^{\left(-1\right)}$ and $\nabla^{\left(1\right)}$ are two special connections on S with respect to the mixture family and the exponential family, respectively. Moreover, they are related by the duality condition, and the following 1-parameter family of connections is defined.

**Definition** **10**($\nabla^{\left(\alpha \right)}$-connection)**.**
*For α∈R, the $\nabla^{\left(\alpha \right)}$-connection on the statistical model S is defined as*
(18)$$\nabla^{\left(\alpha \right)}=\frac{1+\alpha }{2}\nabla^{\left(1\right)}+\frac{1-\alpha }{2}\nabla^{\left(-1\right)}.$$

**Proposition** **6.**
*The components $\Gamma_{ij,k}^{\left(\alpha \right)}$ can be written as*
(19)$$\Gamma_{ij,k}^{\left(\alpha \right)}=E_{\xi }\left[\left(\partial_{i}\partial_{j}\ell +\frac{1-\alpha }{2}\partial_{i}\ell \partial_{j}\ell \right)\partial_{k}\ell \right].$$


The α-coordinate system associated with the $\nabla^{\left(\alpha \right)}$-connection is endowed with the α-geodesic, which is a straight line on the corresponding coordinates system. Then, we introduce some relevant notions.

**Definition** **11**(*α*-divergence [33]). *Let α be a real parameter. The α-divergence between two probability vectors p and q is defined as*
(20)$$D_{\alpha }\left[p∥q\right]=\frac{4}{1-\alpha^{2}}\left(1-\sum_{i}p_{i}^{\frac{1-\alpha }{2}}q_{i}^{\frac{1+\alpha }{2}}\right).$$

The KL-divergence, which is a special case with α=1, induces the linear connection $\nabla^{\left(1\right)}$ as follows.

**Proposition** **7.**
*The diagonal part of the third mixed derivatives of the KL-divergence is the negative of the Christoffel symbol:*
(21)$$-\partial_{\xi^{i}}\partial_{\xi^{j}}\partial_{\xi_{0}^{k}}D_{KL}\left[p_{\xi_{0}}∥p_{\xi }\right]{|}_{\xi =\xi_{0}}=\Gamma_{ij,k}^{\left(1\right)}\left(\xi_{0}\right).$$


**Proof.** The second derivative in the argument ξ is given by
$$\partial_{\xi^{i}}\partial_{\xi^{j}}D_{KL}\left[p_{\xi_{0}}∥p_{\xi }\right]=-\int_{X}p_{\xi_{0}}\left(x\right)\partial_{\xi^{i}}\partial_{\xi^{j}}\ell_{x}\left(\xi \right)dx,$$
and differentiating it with respect to $\xi_{0}^{k}$ yields
$$\begin{array}{cc} -\partial_{\xi^{i}}\partial_{\xi^{j}}\partial_{\xi_{0}^{k}}D_{KL}\left[p_{\xi_{0}}∥p_{\xi }\right] & =\partial_{\xi_{0}^{k}}\int_{X}p_{\xi_{0}}\left(x\right)\partial_{\xi^{i}}\partial_{\xi^{j}}\ell_{x}\left(\xi \right)dx\\ \; & =\int_{X}p_{\xi_{0}}\left(x\right)\partial_{\xi^{i}}\partial_{\xi^{j}}\ell_{x}\left(\xi \right)\partial_{\xi_{0}^{k}}\ell_{x}\left(\xi \right)dx. \end{array}$$Then, considering the diagonal part, one yields
$$\begin{array}{cc} -\partial_{\xi^{i}}\partial_{\xi^{j}}\partial_{\xi_{0}^{k}}D_{KL}\left[p_{\xi_{0}}∥p_{\xi }\right]{|}_{\xi =\xi_{0}} & =E_{\xi_{0}}\left[\partial_{i}\partial_{j}\ell \left(\xi \right)\partial_{k}\ell \left(\xi \right)\right]\\ \; & =\Gamma_{ij,k}^{\left(1\right)}\left(\xi_{0}\right). \end{array}$$ □

More generally, the α-divergence with α∈R induces the $\nabla^{\left(\alpha \right)}$-connection.

**Definition** **12**(*α*-representation [34]). *For some positive measure $m_{i}^{\frac{1-\alpha }{2}}$, the coordinate system $\theta =(\theta^{i})$ derived from the α-divergence is*
(22)$$\theta^{i}=m_{i}^{\frac{1-\alpha }{2}}=f_{\alpha }\left(m_{i}\right)$$
*and $\theta^{i}$ is called the α-representation of a positive measure $m_{i}^{\frac{1-\alpha }{2}}$.*

**Definition** **13**(*α*-geodesic [28])**.**
*The α-geodesic connecting two probability vectors p(x) and q(x) is defined as*
(23)$$r_{i}\left(t\right)=c\left(t\right)f_{\alpha }^{-1}\left\{\left(1-t\right)f_{\alpha }\left(p\left(x_{i}\right)\right)+tf_{\alpha }\left(q\left(x_{i}\right)\right)\right\},\;t\in \left[0,1\right]$$
*where c(t) is determined as*
(24)$$c\left(t\right)=\frac{1}{\sum_{i=1}^{n}r_{i}\left(t\right)}.$$

It is known that the appropriate reparameterizations for the parameter *t* are necessary for a rigorous discussion in the space of probability measures [35,36]. However, as mentioned in the literature [35], an explicit expression for the reparametrizations $\tau_{p,a}$ and $\tau_{p,q}$ is unknown. A similar discussion has been made in the derivation of the $\phi_{\beta }$-path [37], where it is mentioned that the normalizing factor is unknown in general. Furthermore, the *f*-mean is not convex depending on the α. For these reasons, it is generally difficult to discuss α-geodesics in probability measures by normalization or reparameterization, and to avoid unnecessary complexity, the parameter *t* is assumed to be appropriately reparameterized.

Let $\psi_{\alpha }\left(\theta \right)=\frac{1-\alpha }{2}\sum_{i=1}^{n}m_{i}$. Then, the dual coordinate system η is given by $\eta =\nabla \psi_{\alpha }\left(\theta \right)$ as
(25)$$\eta_{i}={\left(\theta^{i}\right)}^{\frac{1+\alpha }{1-\alpha }}=f_{-\alpha }\left(m_{i}\right).$$

Hence, it is the (−α)-representation of $m_{i}$.

### 2.2. Generalization of Skew Divergences

From Definition 13, the *f*-interpoloation is considered as an unnormalized version of the α-geodesic. Using the notion of geodesics, skew divergence is generalized in terms of information geometry as follows.

**Definition** **14**(*α*-Geodesical Skew Divergence)**.**
*The α-geodesical skew divergence $D_{GS}^{\left(\alpha ,\lambda \right)}:\;P\times P\to \left[0,\infty \right]$ is defined between two Radon–Nikodym densities p and q of μ-absolutely continuous probability measures by:*
(26)$$\begin{array}{cc} D_{GS}^{\left(\alpha ,\lambda \right)}\left[p∥q\right] & :=D_{KL}\left[p∥m_{f}^{\left(\lambda ,\alpha \right)}\left(p,q\right)\right]\\ \; & =\int_{X}p\ln\frac{p}{m_{f}^{\left(\lambda ,\alpha \right)}\left(p,q\right)}d\mu , \end{array}$$
*where α∈R and λ∈[0,1].*

Some special cases of α-geodesical skew divergence are listed below:$$\begin{array}{cc} (\forall \alpha \in R,\lambda =1)\; & D_{GS}^{\left(\alpha ,1\right)}\left[p∥q\right]=D_{KL}\left[p∥q\right]\\ (\forall \alpha \in R,\lambda =0)\; & D_{GS}^{\left(\alpha ,0\right)}\left[p∥q\right]=D_{KL}\left[p∥p\right]=0\\ (\alpha =1,\forall \lambda \in [0,1])\; & D_{GS}^{\left(1,\lambda \right)}\left[p∥q\right]=\lambda D_{KL}\left[p∥q\right]\;\;\left(\mathrm{scaled}\;\text{KL-divergence}\right)\\ (\alpha =-1,\forall \lambda \in [0,1])\; & D_{GS}^{\left(-1,\lambda \right)}\left[p∥q\right]=D_{S}^{\left(\lambda \right)}\left[p∥q\right]\;\;\left(\mathrm{skew}\;\mathrm{divergence}\right)\\ (\alpha =0,\forall \lambda \in [0,1])\; & D_{GS}^{\left(0,\lambda \right)}\left[p∥q\right]=\int_{X}p\ln\frac{p}{{\left\{\left(1-\lambda \right)\sqrt{p}+\lambda \sqrt{q}\right\}}^{2}}d\mu \\ (\alpha =3,\forall \lambda \in [0,1])\; & D_{GS}^{\left(3,\lambda \right)}\left[p∥q\right]=D_{S}^{\left(\lambda \right)}\left[p∥q\right]+H\left(p\right)+H\left(q\right)\\ (\alpha =\infty ,\forall \lambda \in [0,1])\; & D_{GS}^{\left(\infty ,\lambda \right)}\left[p∥q\right]=\int_{X}p\ln\frac{p}{\min\{p,q\}}d\mu \\ (\alpha =-\infty ,\forall \lambda \in [0,1])\; & D_{GS}^{\left(-\infty ,\lambda \right)}\left[p∥q\right]=\int_{X}p\ln\frac{p}{\max\{p,q\}}d\mu \end{array}$$

Furthermore, α-geodesical skew divergence is a special form of the generalized skew K-divergence [10,38], which is a family of abstract means-based divergences. In this paper, the skew K-divergence touched upon in [10] is characterized in terms of α-geodesic on positive measures, and its geometric and functional analytic properties are investigated. When the Kolmogorov–Nagumo average (i.e., when the function $f^{-1}$ in Equation (Equation 8) is a strictly monotone convex function) the geodesic has been shown to be well-defined [37].

### 2.3. Symmetrization of α-Geodesical Skew Divergence

It is easy to symmetrize the α-geodesical skew divergence as follows.

**Definition** **15**(Symmetrized *α*-Geodesical Skew Divergence)**.**
*The symmetrized α-geodesical skew divergence ${\bar{D}}_{GS}^{\left(\alpha ,\lambda \right)}:P\times P\to \left[0,\infty \right]$ is defined between two Radon–Nikodym densities p and q of μ-absolutely continuous probability measures by:*
(27)$$\begin{array}{cc} {\bar{D}}_{GS}^{\left(\alpha ,\lambda \right)}\left[p∥q\right] & :=\frac{1}{2}\left(D_{GS}^{\left(\alpha ,\lambda \right)}\left[p∥q\right]+D_{GS}^{\left(\alpha ,\lambda \right)}\left[q∥p\right]\right), \end{array}$$
*where α∈R and λ∈[0,1].*

It is seen that ${\bar{D}}_{GS}^{\left(\alpha ,\lambda \right)}\left[p∥q\right]$ includes several symmetrized divergences.
$$\begin{array}{cc} {\bar{D}}_{GS}^{\left(\alpha ,1\right)}\left[p∥q\right] & =\frac{1}{2}\left(D_{KL}\left[p∥q\right]+D_{KL}\left[q∥p\right]\right),\;\;\left(\mathrm{half}\;\mathrm{of}\;\mathrm{Jeffreys}\;\mathrm{divergence}\right)\\ {\bar{D}}_{GS}^{\left(-1,\frac{1}{2}\right)}\left[p∥q\right] & =\frac{1}{2}\left(D_{KL}\left[p∥\frac{p+q}{2}\right]+D_{KL}\left[q∥\frac{p+q}{2}\right]\right),\;\;\left(\text{JS-divergence}\right)\\ {\bar{D}}_{GS}^{\left(-1,\lambda \right)}\left[p∥q\right] & =\frac{1}{2}\left(D_{KL}\left[p∥(1-\lambda )p+\lambda q\right]+D_{KL}\left[q∥(1-\lambda )q+\lambda p\right]\right). \end{array}$$

The last one is the λ-JS-divergence [39], which is a generalization of the JS-divergence.

## 3. Properties of *α*-Geodesical Skew Divergence

In this section, the properties of the α-geodesical skew divergence are studied.

**Proposition** **8**(Non-negativity of the *α*-geodesical skew divergence)**.**
*For α≥−1 and λ∈[0,1], the α-geodesical skew divergence $D_{GS}^{\left(\alpha ,\lambda \right)}\left[p∥q\right]$ satisfies the following inequality:*
(28)$$D_{GS}^{\left(\alpha ,\lambda \right)}\left[p∥q\right]\geq 0.$$

**Proof.** 
*When λ is fixed, the f-interpolation has the following inverse monotonicity with respect to α:*
(29)$$m_{f}^{\left(\lambda ,\alpha \right)}\left(p,q\right)\geq m_{f}^{\left(\lambda ,\alpha^{\prime }\right)}\left(p,q\right),\;\left(\alpha \leq \alpha^{\prime }\right).$$
*From Gibbs’ inequality [40] and Equation (Equation 29), one obtains*$$\begin{array}{cc} D_{GS}^{\left(\alpha ,\lambda \right)}\left[p∥q\right] & =\int_{X}p\ln\frac{p}{m_{f}^{\left(\alpha ,\lambda \right)}\left(p,q\right)}d\mu \\ \; & \geq \left(\int_{X}pd\mu \right)\ln\frac{p}{m_{f}^{\left(\alpha ,\lambda \right)}\left(p,q\right)}\\ \; & \geq 1\cdot \ln1=0. \end{array}$$ □

**Proposition** **9**(Asymmetry of the *α*-geodesical skew divergence)**.**
*α-Geodesical skew divergence is not symmetric in general:*
(30)$$D_{GS}^{\left(\alpha ,\lambda \right)}\left[p|q\right]\neq D_{GS}^{\left(\alpha ,\lambda \right)}\left[q∥p\right].$$

**Proof.** For example, if λ=1, then ∀α∈R, it holds that
$$\begin{array}{cc} D_{GS}^{\left(\alpha ,1\right)}\left[p∥q\right]-D_{GS}^{\left(\alpha ,1\right)}\left[q∥p\right] & =D_{KL}\left[p∥q\right]-D_{KL}\left[q∥p\right], \end{array}$$
and the asymmetry of the KL-divergence results in an asymmetry of the geodesic skew divergence. □

When a function f(x) of x∈[0,1] satisfies f(x)=f(1−x), it is referred to as centrosymmetric.

**Proposition** **10**(Non-centrosymmetricity of the *α*-geodesical skew divergence with respect to *λ*)**.**
*α-Geodesical skew divergence is not centrosymmetric in general with respect to the parameter λ∈[0,1]:*
(31)$$D_{GS}^{\left(\alpha ,\lambda \right)}\left[p∥q\right]\neq D_{GS}^{\left(\alpha ,1-\lambda \right)}\left[p∥q\right].$$

**Proof.** For example, if λ=1, then ∀α∈R, we have
(32)$$\begin{array}{cc} D_{GS}^{\left(\alpha ,\lambda \right)}\left[p∥q\right]-D_{GS}^{\left(\alpha ,1-\lambda \right)}\left[p∥q\right] & =D_{GS}^{\left(\alpha ,1\right)}\left[p∥q\right]-D_{GS}^{\left(\alpha ,0\right)}\left[p∥q\right]\\ \; & =\int_{X}p\ln\frac{p}{q}-\int_{X}p\ln\frac{p}{p}\\ \; & =\int_{X}p\ln\frac{p}{q}\geq 0. \end{array}$$ □

**Proposition** **11**(Monotonicity of the *α*-geodesical skew divergence with respect to *α*)**.**
*α-Geodesical skew divergence satisfies the following inequality for all α∈R,λ∈[0,1].*
$$D_{GS}^{\left(\alpha ,\lambda \right)}\left[p∥q\right]\geq D_{GS}^{\left(\alpha^{\prime },\lambda \right)}\left[p∥q\right],\;\left(\alpha \geq \alpha^{\prime }\right).$$

**Proof.** Obvious from the inverse monotonicity of the *f*-interpolation (Equation 29) and the monotonicity of the logarithmic function. □

Figure 1 shows the monotonicity of the geodesic skew divergence with respect to α. In this figure, divergence is calculated between two binomial distributions.

**Proposition** **12**(Subadditivity of the *α*-geodesical skew divergence with respect to *α*)**.**
*α-Geodesical skew divergence satisfies the following inequality for all α,β∈R,λ∈[0,1]*

$$D_{GS}^{\left(\alpha +\beta ,\lambda \right)}\left[p∥q\right]\leq D_{GS}^{\left(\alpha ,\lambda \right)}\left[p∥q\right]+D_{GS}^{\left(\beta ,\lambda \right)}\left[p∥q\right].$$

**Proof.** For some α and λ, $m_{f}^{\left(\lambda ,\alpha \right)}$ takes the form of the Kolmogorov mean [29], which is obvious from its continuity, monotonicity and self-distributivity. □

**Proposition** **13**(Continuity of the *α*-geodesical skew divergence with respect to *α* and *λ*)**.**
*α-Geodesical skew divergence has the continuity property.*

**Proof.** We can prove from the continuity of the KL-divergence and the Kolmogorov mean. □

Figure 2 shows the continuity of the geodesic skew divergence with respect to α and λ. Both source and target distributions are binomial distributions. From this figure, it can be seen that the divergence changes smoothly as the parameters change.

**Lemma** **1.**
*Suppose α→∞. Then,*
(33)$$\lim_{\alpha \to \infty }D_{GS}^{\left(\alpha ,\lambda \right)}\left[p∥q\right]=\int_{X}p\ln\frac{p}{\min\left\{p,q\right\}}d\mu$$
*holds for all λ∈[0,1].*


**Proof.** Let $u=\frac{1-\alpha }{2}$. Then $\lim_{\alpha \to \infty }u=-\infty$. Assuming $p_{0}\leq p_{1}$, it holds that
$$\begin{array}{cc} \lim_{\alpha \to \infty }m_{f}^{\left(\lambda ,\alpha \right)}\left(p_{0},p_{1}\right) & =\lim_{u\to -\infty }{\left(\left(1-\lambda \right)p_{0}^{u}+\lambda p_{1}^{u}\right)}^{\frac{1}{u}}\\ \; & =p_{0}\lim_{u\to -\infty }{\left(\left(1-\lambda \right)+\lambda {\left(\frac{p_{1}}{p_{0}}\right)}^{u}\right)}^{\frac{1}{u}}\\ \; & =p_{0}=\min\left\{p_{0},p_{1}\right\}. \end{array}$$Then, the following equality
$$\begin{array}{cc} \lim_{\alpha \to \infty }D_{GS}^{\left(\alpha ,\lambda \right)}\left[p∥q\right] & =\int_{X}p\ln\frac{p}{\lim_{\alpha \to \infty }m_{f}^{\left(\lambda ,\alpha \right)}\left(p_{0},p_{1}\right)}d\mu \\ \; & =\int_{X}p\ln\frac{p}{\min\left\{p,q\right\}}d\mu \end{array}$$
holds. □

**Lemma** **2.**
*Suppose α→−∞. Then,*
(34)$$\lim_{\alpha \to \infty }D_{GS}^{\left(\alpha ,\lambda \right)}\left[p∥q\right]=\int_{X}p\ln\frac{p}{\max\left\{p,q\right\}}d\mu$$
*holds for all λ∈[0,1].*


**Proof.** Let $u=\frac{1-\alpha }{2}$. Then $\lim_{\alpha \to -\infty }u=\infty$. Assuming $p_{0}\leq p_{1}$, it holds that
$$\begin{array}{cc} \lim_{\alpha \to \infty }m_{f}^{\left(\lambda ,\alpha \right)}\left(p_{0},p_{1}\right) & =\lim_{u\to -\infty }{\left(\left(1-\lambda \right)p_{0}^{u}+\lambda p_{1}^{u}\right)}^{\frac{1}{u}}\\ \; & =p_{1}\lim_{u\to -\infty }{\left(\left(1-\lambda \right){\left(\frac{p_{0}}{p_{1}}\right)}^{u}+\lambda \right)}^{\frac{1}{u}}\\ \; & =p_{1}=\max\left\{p_{0},p_{1}\right\}. \end{array}$$Then, the following equality
$$\begin{array}{cc} \lim_{\alpha \to -\infty }D_{GS}^{\left(\alpha ,\lambda \right)}\left[p∥q\right] & =\int_{X}p\ln\frac{p}{\lim_{\alpha \to -\infty }m_{f}^{\left(\lambda ,\alpha \right)}\left(p_{0},p_{1}\right)}d\mu \\ \; & =\int_{X}p\ln\frac{p}{\max\left\{p,q\right\}}d\mu \end{array}$$
holds. □

**Proposition** **14**(Lower bound of the *α*-geodesical skew divergence)**.**
*α-Geodesical skew divergence satisfies the following inequality for all α∈R,λ∈[0,1].*
(35)$$D_{GS}^{\left(\alpha ,\lambda \right)}\left[p∥q\right]\geq \int_{X}p\ln\frac{p}{\max\{p,q\}}d\mu .$$

**Proof.** It follows from the definition of the inverse monotonicity of *f*-interpolation (Equation 29) and Lemma 2. □

**Proposition** **15**(Upper bound of the *α*-geodesical skew divergence)**.**
*α-Geodesical skew divergence satisfies the following inequality for all α∈R,λ∈[0,1].*
(36)$$D_{GS}^{\left(\alpha ,\lambda \right)}\left[p∥q\right]\leq \int_{X}p\ln\frac{p}{\min\{p,q\}}d\mu .$$

**Proof.** It follows from the definition of the *f*-interpolation (Equation 29) and Lemma 1. □

**Theorem** **1**(Strong convexity of the *α*-geodesical skew divergence)**.**
*α-Geodesical skew divergence $D_{GS}^{\left(\alpha ,\lambda \right)}\left[p∥q\right]$ is strongly convex in p with respect to the total variation norm.*

**Proof.** Let $r:=m_{f}^{\left(\alpha ,\lambda \right)}\left(p,q\right)$ and $f_{j}:=\frac{p_{j}}{r}\;\left(j=0,1\right)$, so that $f_{t}=\frac{p_{t}}{r}\;\left(t\in \left(0,1\right)\right)$. From Taylor’s theorem, for g(x):=xlnx and j=0,1, it holds that
$$\begin{array}{c} g\left(f_{j}\right)=g\left(f_{t}\right)+g^{\prime }\left(f_{t}\right)\left(f_{j}-f_{t}\right)+{\left(f_{j}-f_{t}\right)}^{2}\int_{0}^{1}g^{\prime \prime }\left(\left(1-s\right)f_{t}+sf_{j}\right)\left(1-s\right)ds. \end{array}$$Let
$$\begin{array}{cc} \delta & :=\left(1-t\right)g\left(f_{0}\right)+tg\left(f_{1}\right)-g\left(f_{t}\right)\\ \; & =\left(1-t\right)t{\left(f_{1}-f_{0}\right)}^{2}\int_{0}^{1}\left(\frac{t}{\left(1-s\right)f_{t}+sf_{0}}+\frac{1-t}{\left(1-s\right)f_{t}+sf_{1}}\right)\left(1-s\right)ds\\ \; & =\left(1-t\right)t{\left(f_{1}-f_{0}\right)}^{2}\int_{0}^{1}\left(\frac{t}{f_{u_{0}}\left(t,s\right)}+\frac{1-t}{f_{u_{1}}\left(t,s\right)}\right)\left(1-s\right)ds, \end{array}$$
where
$$\begin{array}{cc} u_{j}\left(t,s\right):= & (1-s)t+jt,\\ f_{\mu_{j}}\left(t,s\right):= & \left(1-s\right)f_{t}+sf_{j}. \end{array}$$Then,
$$\begin{array}{cc} \Delta & :=\left(1-t\right)H\left(p_{0}\right)+tH\left(p_{1}\right)-H\left(p_{t}\right)\\ \; & =\int \delta dr\\ \; & =\left(1-t\right)t\int_{0}^{1}\left(1-s\right)ds\left[tI\left(u_{0}\left(t,s\right)\right)+\left(1-t\right)I\left(u_{1}\left(t,s\right)\right)\right], \end{array}$$
where
$$\begin{array}{cc} ∥p_{1}-p_{0}∥:= & \int |dp_{1}-dp_{0}|d\mu ,\\ H(p):= & D_{GS}^{\left(\alpha ,\lambda \right)}\left[p∥r\right]=\int p\ln\frac{p}{r}d\mu ,\\ I(u):= & \int \frac{{\left(f_{1}-f_{0}\right)}^{2}}{f_{u}}dr. \end{array}$$Now, it is suffice to prove that $\Delta \geq \frac{t(1-t)}{2}{∥p_{1}-p_{0}∥}^{2}$. For all u∈(0,1), it is seen that $p_{1}$ is absolutely continuous with respect to $p_{u}$. Let $g_{u}:=\frac{p_{1}}{p_{u}}=\frac{f_{1}}{f_{u}}$. One obtains
$$\begin{array}{cc} I(u) & =\frac{1}{{\left(1-u\right)}^{2}}\int \frac{{\left(f_{1}-f_{u}\right)}^{2}}{f_{u}}dr\\ \; & =\frac{1}{{\left(1-u\right)}^{2}}\int {\left(g_{u}-1\right)}^{2}dp_{u}\\ \; & \geq \frac{1}{{\left(1-u\right)}^{2}}{\left(\int |g_{u}-1|dp_{u}\right)}^{2}\\ \; & =\frac{1}{{\left(1-u\right)}^{2}}∥p_{1}-p_{u}{∥}^{2}={∥p_{1}-p_{0}∥}^{2}, \end{array}$$
and hence, for j=0,1,
$$\Delta \geq \frac{t(1-t)}{2}{∥p_{1}-p_{0}∥}^{2}.$$ □

## 4. Natural *α*-Geodesical Skew Divergence for Exponential Family

In this section, the exponential family is considered in which the probability density function is given by
(37)$$p\left(x;\theta \right)=\exp\left\{\theta \cdot x+k(x)-\psi (\theta )\right\},$$
where x is a random variable. In the above equation, $\theta =(\theta^{1},\cdots ,\theta^{n})$ is an *n*-dimensional vector parameter to specify distribution, k(x) is a function of x and ψ corresponds to the normalization factor.

In skew divergence, the probability distribution of the target is a weighted average of the two distributions. This implicitly assumes that interpolation of the two probability distributions is properly given by linear interpolation. Here, in the exponential family, the interpolation between natural parameters rather than interpolation between probability distributions themselves is considered. Namely, the geodesic connecting two distributions $p(x;\theta_{p})$ and $q(x;\theta_{q})$ on the θ-coordinate system is considered:(38)$$\theta \left(\lambda \right)=\left(1-\lambda \right)\theta_{p}+\lambda \theta_{q},$$
where λ∈[0,1] is the parameter. The probability distributions on the geodesic θ(λ) are
(39)$$\begin{array}{cc} p(x;\lambda ) & =p(x;\theta (\lambda ))\\ \; & =\exp\left\{\lambda \left(\theta_{q}-\theta_{p}\right)\cdot x+\theta_{p}\cdot x-\psi \left(\lambda \right)\right\}. \end{array}$$

Hence, a geodesic itself is a one-dimensional exponential family, where λ is the natural parameter. A geodesic consists of a linear interpolation of the two distributions in the logarithmic scale because
(40)$$\ln p\left(x;\lambda \right)=\left(1-\lambda \right)\ln p(x;\theta_{p})+\lambda \ln p\left(x;\theta_{q}\right)-\psi \left(\lambda \right).$$

This corresponds to the case α=1 on the *f*-interpolation with normalization factor c(λ)=exp{−ψ(λ)},
(41)$$p\left(x;\theta \left(\lambda \right)\right)=m_{f}^{\left(\lambda ,1\right)}\left(p\left(x;\theta_{p}\right),p\left(x;\theta_{q}\right)\right).$$

This induces the natural geodesic skew divergence with α=1 as
$$\begin{array}{cc} D_{GS}^{\left(1,\lambda \right)}\left[p∥q\right] & =\int_{X}p\ln(\frac{p}{m_{f}^{\left(\lambda ,1\right)}\left(p,q\right)})d\mu \\ \; & =\int_{X}p\ln p-p\ln\left(m_{f}^{\left(\lambda ,1\right)}\left(p,q\right)\right)d\mu \\ \; & =\int_{X}p\ln p-p\ln\left(\exp\{(1-\lambda )\ln p+\lambda \ln q\}\right)d\mu \\ \; & =\int_{X}\left(p\ln p-(1-\lambda )p\ln p-\lambda p\ln q\right)d\mu \\ \; & =\int_{X}\left(\lambda p\ln p-\lambda p\ln q\right)d\mu \\ \; & =\lambda \int_{X}p\ln\frac{p}{q}d\mu \\ \; & =\lambda D_{KL}\left[p∥q\right], \end{array}$$
and this is equal to the scaled KL divergence.

More generally, let $\theta_{P}^{\left(\alpha \right)}$ and $\theta_{Q}^{\left(\alpha \right)}$ be the parameter representations on the α-coordinate system of probability distributions *P* and *Q*. Then, the geodesics between them are represented as in Figure 3, and it induces the α-geodesical skew divergence.

## 5. Function Space Associated with the *α*-Geodesical Skew Divergence

To discuss the functional nature of the α-geodesical skew divergence in more depth, the function space it constitutes is considered. For an α-geodesical skew divergence $f_{q}^{\left(\alpha ,\lambda \right)}\left(p\right)=D_{GS}^{\left(\alpha ,\lambda \right)}\left[p∥q\right]$ with one side of the distribution fixed, let the entire set be
(42)$$F_{q}=\left\{f_{q}^{\left(\alpha ,\lambda \right)}|\alpha \in R,\lambda \in \left[0,1\right]\right\}.$$

For $f_{q}^{\left(\alpha ,\lambda \right)}\in F_{q}$, its semi-norm is defined by
(43)$$∥f_{q}^{\left(\alpha ,\lambda \right)}{∥}_{p}:=\int_{X}{\left(|f_{q}^{\left(\alpha ,\lambda \right)}{|}^{p}d\mu \right)}^{\frac{1}{p}}.$$

By defining addition and scalar multiplication for $f_{q}^{\left(\alpha ,\lambda \right)},g_{q}^{\left(\alpha ,\lambda \right)}\in F_{q}$, c∈R as follows, $F_{q}$ becomes a semi-norm vector space: (44)$$\begin{array}{cc} \left(f_{q}^{\left(\alpha ,\lambda \right)}+g_{q}^{\left(\alpha ,\lambda \right)}\right)\left(u\right):= & f_{q}^{\left(\alpha ,\lambda \right)}\left(u\right)+g_{q}^{\left(\alpha ,\lambda \right)}\left(u\right)=D_{GS}^{\left(\alpha ,\lambda \right)}\left[u∥q\right]+D_{GS}^{\left(\alpha^{\prime },\lambda^{\prime }\right)}\left[u∥q\right], \end{array}$$(45)$$\begin{array}{cc} (cf)(u):= & cf_{q}^{\left(\alpha ,\lambda \right)}\left(u\right)=c\cdot D_{GS}^{\left(\alpha ,\lambda \right)}\left[u∥q\right]. \end{array}$$

**Theorem** **2.**
*Let N be the kernel of ${∥\cdot ∥}_{p}$ as follows:*
(46)$${Nker(∥\cdot ∥}_{p})=\left\{f_{q}^{\left(\alpha ,\lambda \right)}|f_{q}^{\left(\alpha ,\lambda \right)}=0\right\}.$$

*Then the quotient space $V(F_{q},∥\cdot {∥}_{p})/N$ is a Banach space.*


**Proof.** It is sufficient to prove that $f_{q}^{\left(\alpha ,\lambda \right)}$ is integrable to the power of *p* and that V is complete. From Proposition 15, the α-geodesical skew divergence is bounded from above for all α∈R and λ∈[0,1]. Since $f_{q}^{\left(\alpha ,\lambda \right)}$ is continuous, we know that it is *p*-power integrable.Let $\left\{f_{n}\right\}$ be a Cauchy sequence of V:
$$\lim_{n,m\to \infty }{∥f_{n}-f_{m}∥}_{p}=0.$$As n(k),k=1,2,⋯, can be taken to be monotonically increasing and
$$∥f_{n}-f_{n(k)}{∥}_{p}<2^{-k}$$
with respect to n>n(k), let
$$∥f_{n(k+1)}-f_{n(k)}{∥}_{p}<2^{-k}.$$If $g_{n}=|f_{n(1)}|+\sum_{j=1}^{n-1}\left|f_{n(j+1)}-f_{n(j)}\right|\in V$, it is non-negatively monotonically increasing at each point, and from the subadditivity of the norm, $∥g_{n}{∥}_{p}\leq {∥f_{n(1)}∥}_{p}+\sum_{j=1}^{n-1}2^{-j}$. From the monotonic convergence theorem, we have
$$∥\lim_{n\to \infty }g_{n}{∥}_{p}=\lim_{n\to \infty }∥g_{n}{∥}_{p}\leq {∥f_{n(1)}∥}_{p}+1<\infty .$$That is, $\lim_{n\to \infty }g_{n}$ exists almost everywhere, and $\lim_{n\to \infty }g_{n}\in V$. From $\lim_{n\to \infty }g_{n}<\infty$, we have
$$f_{n(1)}+\sum_{j=1}^{n-1}\left(f_{n(j+1)}-f_{n(j)}\right)=\lim_{n\to \infty }f_{n(1)}$$
converges absolutely almost everywhere to $|\lim_{n\to \infty }f_{n(n)}|\leq \lim_{n\to \infty }g_{n},a.e.$ That is, $\lim_{n\to \infty }f_{n(n)}\in V$. Then
$$|\lim_{n\to \infty }f_{n}-f_{n(n)}|\leq \lim_{n\to \infty }g_{n}$$
and from the superior convergence theorem, we can obtain
$$\lim_{n\to \infty }∥\lim_{n\to \infty }f_{n}-f_{n(n)}{∥}_{p}=0$$We have now confirmed the completeness of V. □

**Corollary** **1.**
*Let*
(47)$$F_{+}=\left\{f_{q}^{\left(\alpha ,\lambda \right)}|\alpha \in R,\lambda \in \left(0,1\right],q\in P\right\}.$$

*Then the space $V_{+}:=(F_{+}{,∥\cdot ∥}_{p})$ is a Banach space.*


**Proof.** If we restrict λ∈(0,1], $D_{GS}^{\left(\alpha ,\lambda \right)}\left[u∥q\right]=0$ if and only if u=q. Then, $V_{+}$ has the unique identity element, and then $V_{+}$ is a complete norm space. □

Consider the second argument *Q* of $D_{GS}^{\left(\alpha ,\lambda \right)}\left(P||Q\right)$ is fixed, which is referred to as the reference distribution. Figure 4 shows values of the α-geodesical skew divergence for a fixed reference *Q*, where both *P* and *Q* are restricted to be Gaussian. In this figure, the reference distribution is N(0,0.5) and the parameters of input distributions are varied in μ∈[0,4.5] and $\sigma^{2}\in \left[0.5,2.3\right]$. From this figure, one can see that a larger value of α emphasizes the discrepancy between distributions *P* and *Q*. Figure 5 illustrates a coordinate system associated with the α-geodesical skew divergence for different α. As seen from the figure, for the same pair of distributions *P* and *Q*, the value of divergence with α=3 is larger than that with α=−1.

## 6. Conclusions and Discussion

In this paper, a new family of divergence is proposed to address the computational difficulty of KL-divergence. The proposed α-geodesical skew divergence is a natural derivation from the concept of α-geodesics in information geometry and generalizes many existing divergences.

Furthermore, α-geodesical skew divergence leads to several applications. For example, the new divergence can be applied to the annealed importance sampling by the same analogy as in previous studies using q-paths [41]. It could also be applied to linguistics, a field in which skew divergence was originally used [19]. 

## Figures and Tables

**Figure 1 entropy-23-00528-f001:**
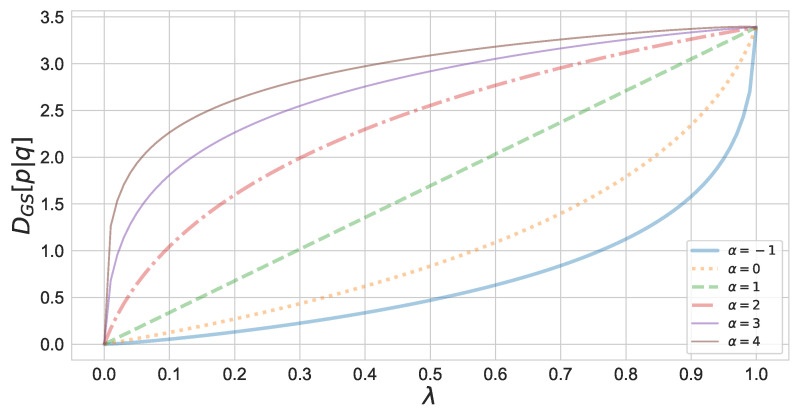
Monotonicity of the α-geodesical skew divergence with respect to α. The α-geodesical skew divergence between the binomial distributions p=B(10,0.3) and q=B(10,0.7) has been calculated.

**Figure 2 entropy-23-00528-f002:**
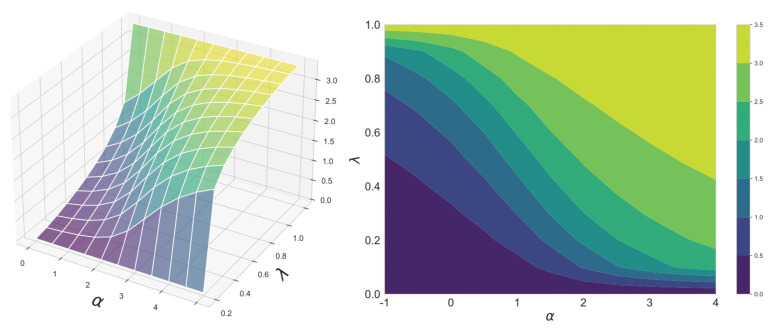
Continuity of the α-geodesmcal skew divergence with respect to α and λ. The α-geodesical skew divergence between the binomial distributions p=B(10,0.3) and q=B(10,0.7) has been calculated.

**Figure 3 entropy-23-00528-f003:**
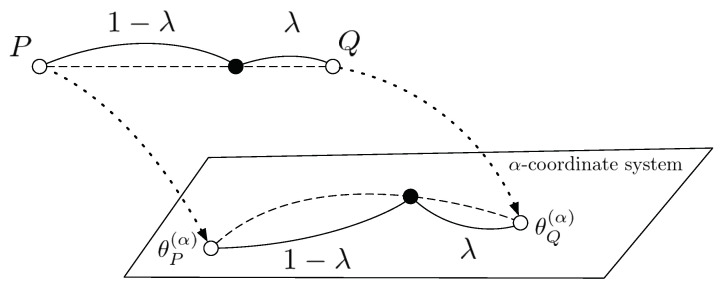
The geodesic between two probability distributions on the α-coordinate system.

**Figure 4 entropy-23-00528-f004:**
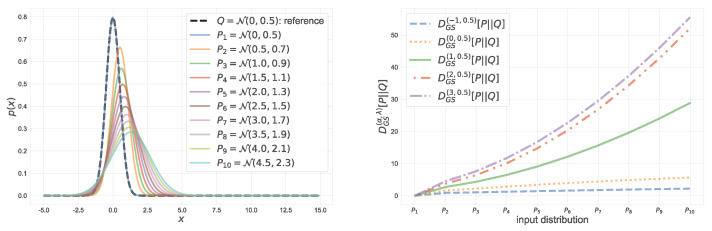
α-geodesical skew divergence between two normal distributions. The reference distribution is Q=N(0,0.5). For $P_{1},P_{2},\ldots ,P_{j},\;\left(j=1,2,\ldots ,10\right)$, let their mean and variance be $\mu_{j}$ and $\sigma_{j}^{2}$, respectively, where $\mu_{j+1}-\mu_{j}=0.5$ and $\sigma_{j+1}^{2}-\sigma_{j}^{2}=0.2$.

**Figure 5 entropy-23-00528-f005:**
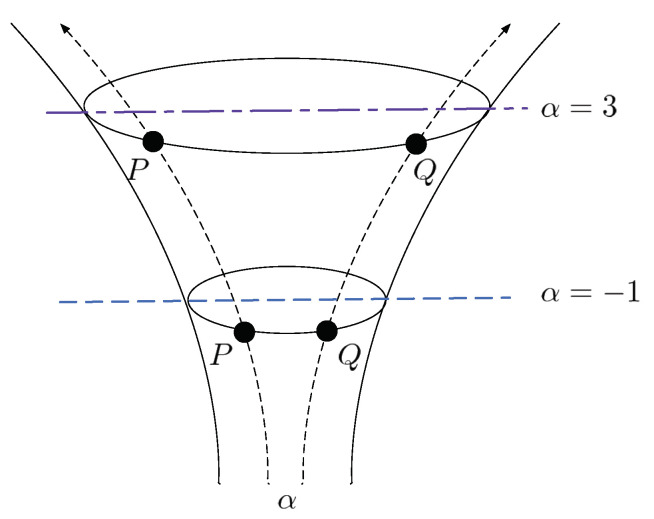
Coordinate system of $F_{q}$ or $F_{+}$. Such a coordinate system is not Euclidean.

## Data Availability

Not applicable.

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
