# Peer review of "α-Geodesical Skew Divergence"

_entropy, 2021, doi:10.3390/e23050528_

Round 1

Reviewer 1 Report

Review of the paper α-Geodesical Skew Divergence

Line 37: in the integral, you have to write the domain ꭓ

Line 71: I think you have to remark at first that you are interested particularly in the Fisher-Rao metric, which is the Riemannian metric induced from the Fisher information matrix, and quote some bibliography

Line 97: in the first line of formula, you forgot the pedices in the Christoffel symbols and in the second member you forgot a sum in the first addend

Line 202: in Taylor’s formula, it is not g’ but g’(ft)

Line 204: in the following formula, it is fuj(t,s) = (1-s)ft + fjs

Line 208: in the formula for I(u), you call hu as gu

Line 212: not “in considered” but “is considered”

Line 230: some parenthesis in the integrals are convenient

Line 240: in the following formulas, f and g need pedices and apices

Line 251: it is not ‖gnp ≤ ‖fn(1)p but ‖gnp ≤ ‖fn(1)p + ∑j=1n-1  2-j

Line 254: the second member of the following formula is only fn(n)

Line 255 and line 256: in the limit, it is not fn but fn(n)

Line 257: say better that we can deduce that also fn has limit in the space

Author Response

Thank you for your helpful comments through reading our manuscript.
You have provided very useful comments for the refinement of the formulas throughout our manuscript.

We have corrected the errors in the formulas, typos, etc. as you suggested.

Thank you again for your careful reading and thoughtful comments.

Reviewer 2 Report

Please, read the enclosed file.

Reviewer 3 Report

I have carefully read the manuscript entropy-1187952.   The Authors proposed an information geometric generalization of the skew divergence called the α-geodesical skew divergence. They study in detail their properties and computational advantages in comparison to the Kulback-Lieber divergence.     The manuscript is rigorously written. In my opinion, it provides an original contribution to the study of information skew divergences. Only comment with respect to the bibliography, perhaps the reference J. Math. Phys. 55, 112202(2014) should be added to the list of references.   From my previous observations, I recommend the manuscript for publication in Entropy.  

Author Response

Thank you for your helpful comments with thorough reading of our manuscript. 

Following your comments, we have added the literature to the list of references.

Thank you again for your careful reading and thoughtful comments.

Round 2

Reviewer 2 Report

I do not have further comments. The paper can be accepted in this form.